# Health effects of desert dust and sand storms: a systematic review and meta-analysis protocol

Aurelio Tobias,[1] Angeliki Karanasiou,[1] Fulvio Amato,[1] Marta Roqué,[2,3] Xavier Querol[1]

[1]Institute of Environmental Assessment and Water Research (IDAEA), Spanish National Research Council (CSIC), Barcelona, Spain
[2]Iberoamerican Cochrane Centre, Barcelona, Spain
[3]Biomedical Research Institute Sant Pau (IIB Sant Pau), CIBER Epidemiology and Public Health (CIBERESP), Barcelona, Spain

**Correspondence to**
Dr Aurelio Tobias;
aurelio.tobias@idaea.csic.es

## ABSTRACT

**Introduction** Desert dust concentrations raise concerns about adverse effects on human health. During the last decade, special attention has been given to mineral dust particles from desert dust and sand storms. However, evidence from previous reviews reported inconclusive results on their health effects and the biological mechanism remains unclear. We aim to systematically synthesise evidence on the health effects of desert dust and sand storms accounting for the relevant desert dust patterns from source areas and emissions, transport and composition.

**Methods an analysis** We will conduct a systematic review that investigated the health effects of desert dust and sand storms in any population. The search will be performed for any eligible studies from previous reviews and selected electronic databases until 2018. Study selection and reporting will follow the Preferred Reporting Items for Systematic Reviews and Meta-Analyses guidelines. Data from individual studies will be extracted using a standardised data extraction form. Quality of the studies will be assessed using a risk of bias tool for environmental exposures developed by experts convened by the WHO. A meta-analysis will be performed by calculating the appropriate effect measures of association for binary and continuous outcomes from individual studies. Subgroup analyses will be performed by geographical areas to account for desert dust patterns.

**Ethics and dissemination** No primary data will be collected. For this reason, no formal ethical approval is required. This systematic review will help to fill the research gaps in the knowledge of desert dust on human health. The results will be disseminated through a WHO peer-reviewed publication and a conference presentation.

**PROSPERO registration number** CRD42018091809

## Strengths and limitations of this study

- ► This systematic review will be the first one to retrieve and evaluate published studies on the health effects of desert dust following a standardised protocol for data collection and reporting of findings.
- ► A meta-analysis will be used to provide evidence of the quantitative estimates of the health effects of desert dust exposures, accounting for relevant desert dust patterns from source areas, emissions, transport and composition.
- ► The diversity of methods to identify dust exposure events and the extensive health effects attributed to desert dust exposures might lead to study designs with different methodological characteristics, which will be carefully considered.
- ► The results of this systematic review will provide evidence to fill the gap in the knowledge of the health effects of desert dust and may help to develop appropriate preventive measures for dust episodes.

episodes is a complex issue. It can increase particulate matter ambient concentrations by supplying large loads of mineral (or crustal) dust, but it can also carry anthropogenic pollutants, previously deposited in the source areas or trapped by the high dust air mass during its atmospheric transport,[5 6] and carry large amounts of microorganisms and toxic biogenic allergens.[7 8]

Research on the health effects of desert dust has increased substantially over the last decade. Although epidemiological studies from affected areas suggest the potential health effects of desert dust, the evidence remains unclear, as well as the biological mechanism. Previously published reviews, systematic or not, reported inconsistent results across studies and geographical regions.[9–12] However, the reviewed studies differed in terms of settings, dust exposure assessment methods, outcomes and lagged exposures examined, and epidemiological study designs evaluated. For this reason, none of the previous reviews could perform

## INTRODUCTION
### Background

Desert dust and sand storms play a significant role in different aspects of weather, climate and atmospheric chemistry and represent a severe hazard to the environment and human health.[1 2] Desert dust has a significant impact on air quality, not only in areas close to the source points or regions but over areas even a few thousands of kilometres distant.[3 4] The air quality influence of dust

a meta-analysis to summarise the health effects of desert dust quantitatively. Thus, a systematic review with a proper standardised protocol is required to provide a better knowledge of health effects and mechanism of toxicity of desert dust.

## Objective

The objective of our systematic review is to summarise the evidence on the health effects of desert dust and sand storms, properly accounting for the relevant desert dust patterns from source areas and emissions, transport and composition. This systematic review will be carried out in the framework of the WHO to inform the update of the WHO Global Air Quality Guidelines (AQGs), coordinated by the WHO Regional Office's European Centre for Environment and Health.

## METHODS

This protocol has been registered in PROSPERO and follows the Preferred Reporting Items for Systematic Review and Meta-Analysis Protocols checklist[13] (online supplementary file 1). Any amendments to the protocol will be tracked and dated in PROSPERO.

## Patient and public involvement

Patients and/or public will not be involved in this study, as this is a systematic review protocol.

## Study eligibility criteria

The WHO established the research question for this systematic review at the third Meeting of the Global Platform on Air Quality and Health (Madrid, Spain, March 2017).

### Population

This systematic review will include any human population, from developed and developing countries, living in urban and in rural areas exposed to desert dust and sand storms. Vulnerable population subgroups to the effects of desert dust and sand storms will be included, such as these with specific pre-existing health conditions (eg, respiratory or cardiovascular diseases), pregnant women, newborns, children or the elderly. Whenever applicable, the considered health effect of exposure to desert dust in these vulnerable subgroups of the population will be assessed separately.

### *Types of exposures*

We will consider short-term and long-term exposure to desert dust characterised by source apportionment, component analysis, particle size and composition, use of weather data and back trajectories, or any other suitable method. The exposure to desert dust should be compared with non-exposure to desert dust or exposure to lower concentrations of desert dust.

### *Types of outcome measures*

We will consider any measurable independent change in the risk of adverse health effects (incidence or prevalence)

related to exposure to desert dust as mortality for all-natural causes and cause-specific (eg, cardiovascular and respiratory) and hospital admissions, emergency room admissions and emergency department specific causes. Moreover, based on a previous evaluation of the human health disorders related to desert dust,[14] we will also consider cardiovascular and respiratory symptoms; lung diseases; coccidiomycosis; meningococcal meningitis; dermatological disorder; and deaths or injuries resulting from transport accidents.

### *Types of studies*

This systematic review will consider all types of relevant studies published in peer-reviewed journals that assess the health effects of desert dust and sand storms. However, the studies will be classified according to their observational or experimental design (eg, quasi-experimental and randomised controlled trials). The observational studies will also be classified at the ecological or individual level (eg, surveys, case–control, cohort studies).

## Information sources

We will systematically conduct a comprehensive literature search. First, we will identify all studies already reported in previous reviews, systematic or not, on the health effects of desert dust and sand storms.[9–12] Next, we will conduct a literature search for studies matching the study eligibility criteria in various databases, including MEDLINE (using PubMed) and EMBASE. Finally, references of identified relevant studies will be scanned to identify additional published data matching the study eligibility criteria. The search strategy has been developed in collaboration with a health sciences librarian specialising in systematic search procedures. We used free-text for each search engine, considering different definitions and keywords for desert dust and sand storms and health outcomes (online supplementary file 2). We will retrieve eligible published studies until 2018 without language restrictions and will translate the non-English abstracts and studies, if necessary.

## Selection of studies

Two reviewers will screen the titles and abstracts independently, record and import search results of their findings in EndNote X7 (Clarivate Analytics). All studies will be classified for inclusion or exclusion based on the eligibility of this systematic review and further checked for duplication. Full-text articles will be retrieved if the study is unclear from the title and abstract, and the relevant studies will be further confirmed for inclusion through full-text review. The third reviewer will resolve discrepancies between reviewers.

## Data extraction

Reviewers will use a standardised data extraction form using Excel spreadsheet (Microsoft Corporation). Here, if a given study reports several health outcomes of interest, each outcome will take one record. Likewise, if a given study reports age or gender-specific effects, these will

also be collected. Moreover, if a given study reports more than one effect estimate for each health outcome (eg, for several lags), the one that the authors favoured most will be extracted. Main conclusion/findings for each study will also be collected as free text. The following characteristics of the included studies will be extracted:

1. Citation details (eg, first author, title, journal and date of publication).
2. Study location (eg, city, country, region), study period, characteristics of the study population (eg, age, gender).
3. Study design and statistical analysis.
4. Method to identify dust events and exposure levels for days with and without dust events.
5. Outcome assessment (eg, cause of mortality, morbidity and other cause-specific outcomes).
6. Confounders measured and adjusted for (eg, meteorological variables and other air pollutants).
7. Effect measures of association, and its 95% CI, for days with and without dust events.

## Quality assessment

We will use a new domain-based risk of bias tool to assess the quality of the evidence across studies for associations between specific air pollutants and adverse health outcomes. The risk of bias tool, currently under development by a group of experts on environmental exposures convened by the WHO Guideline Development Group (GDG) for the AQGs, will adapt the domains according to the needs of the observational studies of exposure that form the body of the evidence in air pollution. It will be revised and adapted to by the systematic review team members, and a member of the WHO GDG will support the application of the tool. Two reviewers will assess the risk of bias independently, and disagreements will be resolved through discussion.

## Data synthesis and statistical analysis

For binary outcomes (eg, mortality and morbidity), the relative risk (RR) will be used as the standard effect measure of association across studies. The HR if reported, may be considered equivalent to RRs. If ORs are reported in a given study and the outcome prevalence is >10% they will be approximated as RRs[15] and similarly if the study design allows for it (eg, time-series and time-stratified case-crossover[16]). Meta-analysis input data will be as the percentage increase in the RR (eg, as %RR = (RR-1) ×100%). When reporting effects of particulate matter for days with and without dust events the %RR will be standardised for a given increment of $10 \mu g/m^3$, assuming a linear exposure–outcome relationship. For continuous outcomes, effect measures such as weighted mean difference (WMD) or standardised mean difference (SMD) may be used. The scale of the available data will be primarily used to determine the choice of effect measure if studies report effect estimates with the same or similar scale (WMD) or when the outcome is measured using different scales (SMD).[15]

The measures of the association from individual studies will be summarised in a meta-analysis estimate, with its 95% CI, based on fixed or random-effects model, depending on the heterogeneity of the analysis.[15 17] Statistical heterogeneity of the effect estimates between studies included in the meta-analysis will be assessed using the $I^2$ statistic, where values of 25%, 50% and 75% will be considered as a low, moderate and high degree of heterogeneity, respectively.[18] We will also report the between-study variance ($\tau^2$) and the Cochran Q test for heterogeneity.[19] All analyses will be done using Stata Statistical Software V.15.

Due to the relevance of the study setting and dust exposure assessment, subgroup analyses will be done by geographical areas, considering studies conducted in Eastern Asia, Europe, Middle the East and other regions (eg, Australia and North America). Subgroup analyses by pre-existing health conditions, age groups (14, 15–64, ≥65 years) and gender (male and female) will also be considered whenever data become available in the individual studies.

## Assessment of reporting biases

Publication bias will be assessed when at least 10 studies are included in the meta-analysis, using Begg's rank correlation test and examined using funnel plots.[20]

## Reporting conclusions

The systematic review will be conducted and reported in accordance to the Preferred Reporting Items for Systematic Review and Meta-Analysis[21] standards, with slight adaptations since these were originally intended for healthcare intervention evaluation. We will base our conclusions only on findings from the quantitative or narrative synthesis of included studies for this systematic review. The quality of the evidence will be assessed using an adaptation for environmental studies, convened by the GDG for the AQGs, of the Grading of Recommendations Assessment (GRADE).[22] Studies that undergo meta-analyses will undergo GRADE assessment, but there might also other studies included in the systematic review that could not be used in the meta-analysis but will be used for developing conclusions. The conclusions from this systematic review will be used to inform the update of the WHO AQGs, coordinated by the WHO Regional Office's European Centre for Environment and Health.

## Ethics and dissemination

Formal ethical approval is not required, as no primary data will be collected. The results of this systematic review will fill the current research gaps in the knowledge on the health effects of desert dust and sand storms and would also help to alert the most sensitive population taking appropriate measures during dust episodes. All findings will be shared and disseminated by WHO in a peer-reviewed publication and conference presentation.

**Acknowledgements** Ivan Solà (Iberoamerican Cochrane Centre) for their guidance in reviewing the search strategy proposed in this protocol.

**Contributors** AT is the guarantor. AT drafted the protocol and registered the protocol in PROSPERO. MR and XQ reviewed and commented on the protocol in PROSPERO. AK, FA, MR and XQ all reviewed and commented on this protocol.

**Funding** The World Health Organization, supported by the Ministry of Foreign Affairs of Norway, has funded this systematic review.

**Competing interests** None declared.

**Patient consent for publication** Not required.

**Provenance and peer review** Not commissioned; externally peer reviewed.

**Data availability statement** There are no data in this work. No data are available.

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
