## [Reviewer comments · BMJ Open]

ARTICLE DETAILS

TITLE (PROVISIONAL)	Health effects of desert dust and sand storms: a systematic review and meta-analysis protocol
AUTHORS	Tobias, Aurelio; Karanasiou, Angeliki; Amato, Fulvio; Roqué, Marta; Querol, Xavier

VERSION 1 - REVIEW

REVIEWER	Takehiro Michikawa Toho University, Japan
REVIEW RETURNED	18-Mar-2019

GENERAL COMMENTS	This manuscript reported a protocol of systematic review and meta-analysis of health effects of desert dust. This review will include published studies from 1999 until 2017. Overall, the paper is well written and the design sounds relevant. I have a few comments that should be considered to address the worth of investigation. 1. Selected electronic databases from 1999 until 2017 Why did the authors decide the starting period in 1999?2. Type of outcome measures As cause-specific mortality or morbidity, it is easy to understand that the authors include respiratory and cardiovascular diseases. However, explanation is required about coccidiomycosis; meningococcal meningitis; and dermatological disorder. In addition, it is unclear whether they include suicide as one of outcomes.3. Data synthesis and statistical analysis (page 8, lines 28-30) "If odds Ratios (OR) are reported in the study and the outcome prevalence is higher than 10%, or the study design allows for it, they will be recalculated as RRs." Please, include information on calculation procedure and reference.4. Data synthesis and statistical analysis (page 10, lines 10-17) Pre-existing health conditions (they stated in the population, page 6, line 37) are not presented in this paragraph. Minor comments
--

	Page6, line 35, desert dust “ands” sand storms → desert dust and sand storms Page 10, line 15, age groups (14, 15-65, 65 years) → (14, 15-64, >=65 years)
--	--

REVIEWER	Pavlos Kassomenos University of Ioannina, Greece
REVIEW RETURNED	03-Apr-2019

GENERAL COMMENTS	A very good and innovative work
---------------------------------

REVIEWER	W. Graham Carlos Indiana University, USA
REVIEW RETURNED	03-Apr-2019

GENERAL COMMENTS	1. Under "quality assessment" the "domain-based risk of bias" tool needs more explanation. How dose it specifically improve quality? A citation (if available) may be helpful here. 2. Consider new development of asthma or RADS as an outcome to look at. Perhaps this is what you mean by respiratory "symptoms" but being more explicit here for known lung diseases (beyond symptoms) would be informative.
---

VERSION 1 – AUTHOR RESPONSE

ANSWER TO REVIEWER: 1

Reviewer Name: Takehiro Michikawa

Institution and Country: Toho Universidty, Japan

Please state any competing interests or state 'None declared': None declared

This manuscript reported a protocol of systematic review and meta-analysis of health effects of desert dust. This review will include published studies from 1999 until 2017. Overall, the paper is well written and the design sounds relevant. I have a few comments that should be considered to address the worth of investigation.

We acknowledge the Reviewer's comments that have allowed improving the contents of the manuscript.

1. Selected electronic databases from 1999 until 2017

Why did the authors decide the starting period in 1999?

The literature search was restricted to 1999 because we noticed from previous reviews that the first paper published on the health effects of desert dust and sand storms was in 1999. However, since we are concerned that previous reviews were not systematic, we have considered it best to conduct a complete literature search, retrieving eligible published studies up to 2018, also following the Editorial. We have updated the text accordingly in the Abstract (page 2) and also in the Information sources section (page 6).

We have also improved the protocol registered in PROSPERO (CRD42018091809) including this change. However, this is not yet reflected because the protocol is now being assessed again by PROSPERO. We hope the suggested change will be reflected shortly.

2. Type of outcome measures

As cause-specific mortality or morbidity, it is easy to understand that the authors include respiratory and cardiovascular diseases. However, explanation is required about coccidiomycosis; meningococcal meningitis; and dermatological disorder. In addition, it is unclear whether they include suicide as one of outcomes.

The specific outcomes (coccidiomycosis, meningococcal meningitis, dermatological disorder, etc.) also considered in the systematic review come 'literally' from the specific requirement of the World Health Organisation, when establishing the research question at the 3rd Meeting of the Global Platform on Air Quality and Health (stated in the first paragraph of the Methods section, page 5), after considering a previously published evaluation of the human health disorders related to desert dust (by Goudie et al. 2014). Moreover, based on this evaluation, suicide is not included as a potential outcome.

We have now clarified this point by updating accordingly the Types of outcome measures sub-section (page 6).

3. Data synthesis and statistical analysis (page 8, lines 28-30)

"If odds Ratios (OR) are reported in the study and the outcome prevalence is higher than 10%, or the study design allows for it, they will be recalculated as RRs." Please, include information on calculation procedure and reference.

As suggested by the Reviewer, we have now clarified this issue using the appropriate wording in the Data synthesis and statistical analysis section (page 8). We will "approximate" ORs as RRs, instead to "recalculate" and we also provided the appropriate references for it.

4. Data synthesis and statistical analysis (page 10, lines 10-17)

Pre-existing health conditions (they stated in the population, page 6, line 37) are not presented in this paragraph.

We agree with the Reviewer. We have now indicated in the Data synthesis and statistical analysis section (page 9), that we will also consider the pre-existing health conditions in the subgroup analysis.

We have also improved the protocol registered in PROSPERO (CRD42018091809) including this change. However, this is not yet reflected because the protocol is now being assessed again by PROSPERO. We hope the suggested change will be reflected shortly.

Minor comments

Page 6, line 35, desert dust “ands” sand storms → desert dust and sand storms

Following the Reviewer suggestion, we have now corrected the misprint.

Page 10, line 15, age groups (14, 15-65, 65 years) → (14, 15-64, >=65 years)

Following the Reviewer suggestion, we have now corrected the misprint clearly defining the age groups for the subgroup analysis.

ANSWER TO REVIEWER: 2

Reviewer Name: Pavlos Kassomenos

Institution and Country: University of Ioannina, Greece

Please state any competing interests or state 'None declared': None declared

A very good and innovative work

We acknowledge the Reviewer's comment.

ANSWER TO REVIEWER: 3

Reviewer Name: W. Graham Carlos

Institution and Country: Indiana University, USA

Please state any competing interests or state 'None declared': None declared

We acknowledge the Reviewer's comments that have allowed improving the contents of the manuscript.

1. Under "quality assessment" the "domain-based risk of bias" tool needs more explanation. How dose it specifically improve quality? A citation (if available) may be helpful here.

As requested by the Reviewer we have included a brief statement on the domain-based risk of bias tool (page 8).

However, the Reviewer should note that at the moment we do not have much information, besides the tool is under development by a group of experts on environmental exposures convened by the WHO Guideline Development Group for the Air Quality Guidelines. For this reason, there is no available reference at the moment. Once it becomes available will be used in all the WHO systematic reviews commissioned to evidence for the update of the Air Quality Guidelines.

2. Consider new development of asthma or RADS as an outcome to look at. Perhaps this is what you mean by respiratory "symptoms" but being more explicit here for known lung diseases (beyond symptoms) would be informative.

The Reviewer is right, we considered Asthma, RADS and others, under respiratory symptoms. As the Reviewer requested we have now been more explicit also including lung diseases beyond respiratory symptoms (page 6).